# Full Skin Equivalent Models for Simulation of Burn Wound Healing, Exploring Skin Regeneration and Cytokine Response

**DOI:** 10.3390/jfb14010029

**Published:** 2023-01-04

**Authors:** Patrick P. G. Mulder, Rajiv S. Raktoe, Marcel Vlig, Anouk Elgersma, Esther Middelkoop, Bouke K. H. L. Boekema

**Affiliations:** 1Preclinical Research, Association of Dutch Burn Centres (ADBC), P.O. Box 1015, 1940 AE Beverwijk, The Netherlands; 2Laboratory of Medical Immunology, Department of Laboratory Medicine, Radboud University Medical Center, Geert Grooteplein Zuid 10, 6525 GA Nijmegen, The Netherlands; 3Department of Plastic, Reconstructive and Hand Surgery, Amsterdam UMC, Vrije Universiteit Amsterdam, De Boelelaan 1118, 1081 HV Amsterdam, The Netherlands; 4Tissue Function and Regeneration, Amsterdam Movement Sciences, De Boelelaan 1105, 1081 HV Amsterdam, The Netherlands

**Keywords:** in vitro model, keratinocytes, fibroblasts, wound repair, skin morphogenesis, burn injury, tissue engineering, cytokine response

## Abstract

Healing of burn injury is a complex process that often leads to the development of functional and aesthetic complications. To study skin regeneration in more detail, organotypic skin models, such as full skin equivalents (FSEs) generated from dermal matrices, can be used. Here, FSEs were generated using de-epidermalized dermis (DED) and collagen matrices MatriDerm^®^ and Mucomaix^®^. Our aim was to validate the MatriDerm- and Mucomaix-based FSEs for the use as in vitro models of wound healing. Therefore, we first characterized the FSEs in terms of skin development and cell proliferation. Proper dermal and epidermal morphogenesis was established in all FSEs and was comparable to ex vivo human skin models. Extension of culture time improved the organization of the epidermal layers and the basement membrane in MatriDerm-based FSE but resulted in rapid degradation of the Mucomaix-based FSE. After applying a standardized burn injury to the models, re-epithelization occurred in the DED- and MatriDerm-based FSEs at 2 weeks after injury, similar to ex vivo human skin. High levels of pro-inflammatory cytokines were present in the culture media of all models, but no significant differences were observed between models. We anticipate that these animal-free in vitro models can facilitate research on skin regeneration and can be used to test therapeutic interventions in a preclinical setting to improve wound healing.

## 1. Introduction

Wound healing of deep and large wounds is often problematic and can lead to medical complications such as hyper-inflammation and excessive scarring of the skin. In turn, these complications can lead to delayed recovery and poor aesthetic outcomes [1,2,3,4]. To improve the treatment of burn injuries, the processes underlying skin regeneration need to be better understood. Furthermore, there is a need for appropriate in vitro models to facilitate drug discovery and testing [5].

Research on cellular processes in burn wound healing is generally performed on animals [6,7,8]. Translation to the human situation is, however, difficult due to physiological differences between animals and humans [9,10]. In addition, in our modern society we strive for innovative, animal-free ways of conducting research [11]. Human studies, on the other hand, are limited by the absence of baseline values, heterogeneity among patients, and restrictions in the collection of tissue samples [6,7]. Therefore, there is an important demand for alternative models to study burn injuries.

Organotypic skin models are useful alternatives to animal experimentation and can be used as a research instrument to study defined aspects of skin trauma, based on the behavior of human cells [12,13,14]. Moreover, these models are easily adjustable to study interactions of specific cell types and can be used to evaluate the effect of therapeutic interventions. Scratch models, in which a scratch is made in a monolayer of a single cell type, can be used to study cell migration and proliferation after wounding. However, such scratch models rely on a single type of cell, usually keratinocytes or fibroblasts [15]. Alternatively, 3D culture models can be used to study skin diseases [12,13,14,16,17,18,19]. 3D culture models resemble a more natural and complete environment for cells; however, they are often produced from hydrogels and seeded with immortalized cell lines or animal cells instead of primary isolated human cells. Furthermore, gel-based models are less suitable for the study of thermal trauma because they might not be strong enough to withstand the injury.

More relevant and robust in vitro culture models for the study of wound healing are full skin equivalents produced from dermal scaffolds seeded with fibroblasts and keratinocytes [20,21]. Such FSEs are uniform, as there is less variation in the matrix, and more representative of the in vivo situation than the aforesaid culture models, because collagen is the predominant component [22]. FSEs produced from dermal collagen-elastin scaffolds provide a durable extracellular matrix architecture that supports cell anchorage [23,24]. In this study, we generated FSEs from the dermal substitutes MatriDerm^®^ and Mucomaix^®^. These dermal matrices are clinically used in combination with split thickness autografts to treat full-thickness skin defects and support skin regeneration [25,26,27,28]. Therefore, FSEs generated using these matrices are relevant in vitro study models. We validated the different FSE models by studying skin development, cell differentiation, cell viability, and cytokine production. A standardized thermal contact injury was applied to evaluate in vitro wound healing. We compared the performance of these FSEs to cultured ex vivo human skin and de-epidermalized dermis (DED)-based FSEs.

## 2. Materials and Methods

### 2.1. Isolation of Keratinocytes and Fibroblasts

See Appendix A for the contents of the culture media. Healthy skin samples were obtained from adult patients who underwent elective surgery at the Departments of Surgery or Plastic and Reconstructive Surgery of the Red Cross Hospital. Eleven skin tissue samples were used, originating from abdominal, leg, or arm reconstructions wherein excess skin was removed (donor age: 43.8 ± 11.7 years old; donor sex: 72.7% female). These samples were collected in the period between January 2021 and July 2021. Consent for the use of these anonymized, post-operative residual tissue samples was received through the informed opt-out protocol of the Red Cross Hospital, which was in accordance with the national guidelines (https://www.coreon.org/ accessed on 23 November 2020) and approved by the institutional privacy officers. Subjects were actively informed of this procedure and were able to easily withdraw at any point. Split-thickness samples of 0.3 mm were harvested using a dermatome (Aesculap AG & Co. KG, Tuttlingen, Germany). The epidermis was separated from the dermis using forceps after incubating the harvested skin samples in 0.25% dispase (Gibco) at 37 °C for 45 min. For fibroblast isolation, the dermal part of the split skin was cut into small pieces and submerged into a 0.25% collagenase (Roche)/0.25% dispase solution at 37 °C for 2 h. After addition of 1 mM EDTA/PBS to inhibit collagenase, the cell suspension was poured through a 500 µm cell strainer and centrifuged for 10 min at 360× *g*. The cell pellet was resuspended in culture medium (Appendix A) and poured through a 70 µm cell strainer and cultured at 37 °C and 5% CO_2_. For keratinocyte isolation, the epidermis was transferred into 0.05% trypsin and incubated for 20 min at 37 °C. The cell suspension was poured through a 70 µm cell strainer and centrifuged for 10 min at 110× *g*. Next, the cell pellet was washed in culture medium and centrifuged for 10 min at 160× *g*. The cell pellet was then resuspended in CellnTec-07S culture medium, and keratinocytes were transferred onto a 1 µg/cm^2^ collagen-type-IV-coated culturing flask at 37 °C and 5% CO_2_.

### 2.2. Full Skin Equivalent Models

De-epidermalized dermis (DED; European Tissue Bank BISLIFE, Beverwijk, The Netherlands), MatriDerm^®^ (thickness 3 mm; MedSkin Solutions Dr. Suwelack AG, Billerbeck, Germany), and Mucomaix^®^ (thickness 3 mm; Matricel GmbH, Hertzogenrath, Germany) were cut into square pieces of 1.44 cm^2^. At day one, 200,000 fibroblasts were seeded onto the matrices (for DED, on the reticular side in a metal ring), and the matrices were submerged in culture medium for 4 days at 37 °C and 5% CO_2_. Subsequently, 100,000 keratinocytes (from frozen stock) were seeded on the opposite side (for DED, on the papillary side in a metal ring), and the models were cultured submerged in FSE I medium containing 4 ng/mL KGF and 1 ng/mL EGF for 4 days at 37 °C and 5% CO_2_. Next, the FSEs were transferred to a stainless-steel grid and cultured air-exposed in FSE II medium containing 4 ng/mL KGF and 1 ng/mL EGF. From day 11, FSEs were cultured in FSE III medium containing 2 ng/mL KGF and 0.5 ng/mL EGF and from day 15 onward in FSE III medium that was refreshed twice a week. Cell numbers and culture times are based on our preliminary experiments where we optimized these settings.

### 2.3. Ex Vivo Human Skin Model

Using a dermatome (Aesculap AG & Co. KG, Tuttlingen, Germany), 0.5 mm split-thickness skin was harvested from human skin and cut into square pieces of 1.44 cm^2^. These models were transferred to a stainless-steel grid and cultured air-exposed at 37 °C with 5% CO_2_ in FSE II medium that was refreshed twice a week.

### 2.4. Induction of Burn Injury

A copper plate (2mm × 10 mm) attached to a PACE intelliHeat ST50 soldering iron (PACE, Vass, NC, USA) was heated to 80 °C and applied to the epidermal side of the models for 20 s without exerting pressure. The temperature of the copper device was measured by an external digital thermometer (Farnell InOne, Utrecht, The Netherlands). FSEs were put in culture for 21 days before they were burn-injured and then cultured for 2 h (T0), 1 week, and 2 weeks. In parallel, burn injury was induced on the ex vivo human skin models, and these models were then also cultured for 2 h (T0), 1 week, and 2 weeks. For both FSEs and ex vivo human skin models, the medium was refreshed twice a week. Figure 1 shows a scheme of the experiment.

### 2.5. Immunohistochemistry

Kryofix (50% ethanol, 3% PEG300)-fixed paraffin-embedded (KFPE) samples were cut into 5 µm thick sections and rehydrated followed by hematoxylin and eosin staining or blocking of endogenous peroxidase using 1% hydrogen peroxide for 15 min at RT. After antigen retrieval was performed (Appendix A), sections were pre-incubated with 5% normal goat serum (Abcam, Cambridge, UK) diluted in PBS + 1% bovine serum albumin. Sections were then incubated with primary antibodies for the detection of pan-cytokeratin, cytokeratin-10, cytokeratin-15, cytokeratin-17, involucrin, collagen IV, laminin α 5, vimentin, aSMA, Ki67, and BrdU (Appendix A) for 1 h at RT followed by incubation with a poly-HRP-goat-anti-mouse or rabbit secondary antibody (Bright Vision, VWR, Amsterdam, The Netherlands) for 30 min at RT. After washing, detection was established using 3,3′-Diaminobenzidine (DAB). After DAB staining was completed, sections were counterstained with hematoxylin, dehydrated, and mounted with Eukit Mounting Medium (Sigma-Aldrich, St. Louis, MO, USA). For 5-bromo-2′-deoxyuridine (BrdU) staining, culture medium was supplemented with 20 µM BrdU (Sigma-Aldrich, St. Louis, MO, USA) 24 h before termination.

### 2.6. Microscopy

Microscopic visualization was performed with a Zeiss Axioskop40FL microscope (Zeiss, Breda, The Netherlands). Images were acquired using a Nikon Eclipse TS2 camera and the NIS-Elements software version 4.4 (Nikon Instruments, Amsterdam, The Netherlands).

### 2.7. Re-Epithelization Rate

Re-epithelization length was measured in microscopic images of H&E-stained sections using standardized measurement to calculate µm/pixel in NIS-Elements software. As both sides of each model were measured, the mean was used in the analysis.

### 2.8. Immunoassay of Culture Medium

Cytokines, chemokines, and growth factors were analyzed in samples of culture medium at T0, T + 1–4 days, T + 5–7 days, and T + 8–11 days (after burn injury). Samples from biological duplicates were pooled per donor (n = 3 donors). Neat samples were measured using the Human Essential Immune Response LegendPlex Multi-analyte Flow Assay kit (cat. 740929; BioLegend), according to the manufacturer’s instruction, and were acquired on a flow cytometer (MACS Quant Analyzer 10, Miltenyi Biotec GmbH, Bergisch Gladbach, Germany). This 13-plex immunoassay included: IL-1β, IL-2, IL-4, IL-6, IL-8 (CXCL-8), IL-10, IL-12p70, IL-17A, IP-10 (CXCL10), MCP-1 (CCL2), IFN-γ, TNF-α, and TGF-β1. Concentrations were determined using FlowLogic software (Inivai Technologies, Victoria, Australia) and re-calculated to pg/mL per day of culture to compensate for differences in intervals of medium changes. When cytokine levels were out of range of the standard, either the lowest level of quantification or the highest level of quantification was used.

### 2.9. Statistical Analysis and Data Visualization

Differences in re-epithelization length and cytokine levels between different time points were explored using the Mann–Whitney U tests in Graphpad version 5.01 (PRISM, La Jolla, CA, USA), and only significant differences were shown in the graphs. A *p*-value of <0.05 was considered to be statistically significant. The data were visualized using Graphpad version 5.01 and R (ggplot package, open source).

## 3. Results

### 3.1. Skin Morphogenesis in Full Skin Equivalent Models Was Similar to Ex Vivo Human Skin

FSEs were generated from dermal matrices DED, MatriDerm, and Mucomaix. To validate our FSE models, we first studied skin development after 3 weeks of initial culture (indicated as time T0) and compared this to ex vivo normal human skin (Figure 2A). At T0, the DED-based models contained a completely developed dermis and a pan-cytokeratin-positive epidermis (Figure 2B). At T + 2 weeks, the epidermal and dermal structures developed further and included a thickened stratum corneum that was similar to that of ex vivo human skin. MatriDerm-based FSEs also displayed a well-developed dermis and pan-cytokeratin-positive epidermis that was comparable to the DED-based FSEs and ex vivo human skin. At T + 2 weeks, the dermis remained intact, and the stratum spinosum became thinner, while the stratum corneum grew thicker. Mucomaix-based FSEs developed a complete, pan-cytokeratin-positive epidermis, but its dermis was rather incomplete due to partial degradation and compaction of the matrix. Extension of the culture time improved organization of the epidermal layer and thickening of the stratum corneum but resulted in further degradation of the matrix. 

### 3.2. Epidermal and Dermal Structures Developed Completely in Full Skin Equivalents

Epidermal and dermal development in the FSEs was examined by immunohistochemical analysis. Cytokeratin 15, present in progenitor keratinocytes [29], was consistently expressed in ex vivo human skin and DED-based FSEs from T0 onward (Figure 3A). In both MatriDerm- and Mucomaix-based models, cytokeratin-15-positive cells were present but did not display a well-organized basal layer. MatriDerm- and Mucomaix-based FSEs developed a basement membrane, as was shown by collagen IV and laminin α 5 expression at the dermal–epidermal junction (Figure 3B,C) [30]. Expression of collagen IV and laminin α 5 gradually increased over time, simultaneously with the improvement of the epidermal architecture. Although organization of the basal layer is not optimal, the FSE models produced an epidermis including a basement membrane, stratum spinosum, and stratum corneum that was similar to ex vivo human skin.

Next, the differentiation status of keratinocytes in the FSEs was assessed by determining the presence of early differentiation marker cytokeratin 10 and late differentiation marker involucrin (Appendix A) [31,32]. In ex vivo human skin and DED-based FSEs, cytokeratin 10 was expressed in all suprabasal layers of the epidermis. While in MatriDerm- and Mucomaix-based FSEs, the presence of cytokeratin-10-positive cells at T0 was limited, the expression of cytokeratin 10 at T + 2 weeks was consistent in the suprabasal layer. Late differentiation marker involucrin was expressed in the stratum granulosum in ex vivo human skin from T0 onward. The DED-based FSEs showed suprabasal involucrin expression at T0, which shifted to the stratum granulosum at T + 2 weeks. As for the MatriDerm- and Mucomaix-based FSEs, involucrin was present in all suprabasal layers from T0 onward.

Expression of stress marker cytokeratin 17 was not present at T0 in the ex vivo human skin models (Appendix A). However, when the ex vivo human skin models were cultured for 1 or 2 weeks, cytokeratin 17 expression was upregulated, similar to the FSEs. In FSEs, cytokeratin 17 was displayed in all epidermal layers. Fibroblast distribution was visualized by analyzing the presence of vimentin in cells in the dermal part of the FSEs (Figure 3D). All FSE models showed a fibroblast-populated dermis with a balanced distribution throughout the matrices. Expression of α smooth muscle actin was studied, but it was not detected in any of the FSEs.

### 3.3. Regenerative Capacity of MatriDerm-Based FSEs Was Similar to DED-Based FSEs and Ex Vivo Human Skin at 2 Weeks after Burn Injury

To investigate the ability of the FSEs to function as burn injury models, we subjected the FSEs to a thermal contact injury of 80 °C for 20 s. In preliminary experiments we found that an injury with these settings caused sufficient damage to induce re-epithelization in similar models. Skin morphology and histology of the models were studied at T0, T + 1 week, and T + 2 weeks (Figure 4 and Appendix A). At the macroscopic level, burn injuries presented in a rectangular shape which remained visible throughout the duration of the culture. In all models, destruction and release of the epidermis was clearly visible on the H&E-stained sections at T0. At T + 1 week and T + 2 weeks, re-epithelization of the epidermis was apparent in the ex vivo human skin model and the DED- and MatriDerm-based FSEs. In the Mucomaix-based model, re-epithelization did not take place. Although this model was clearly damaged, formation of a neo-epidermis was not detected.

Next, we validated the regenerative capacity of the models. Therefore, we studied the presence of proliferating cells in the neo-epidermis and quantified the length of the re-epithelized epidermis (Figure 5). Ki67 staining revealed proliferating cells in and nearby the re-epithelized area in DED- and MatriDerm-based FSEs and the ex vivo human skin model (Figure 5A). In Mucomaix-based FSEs, there were hardly any positive cells present. BrdU was used to study 24 h proliferation in the models (Figure 5B). BrdU-positive keratinocytes were present in the newly formed basal layer of the ex vivo human skin model and the DED-based FSE. In the MatriDerm-based FSE, BrdU-positive keratinocytes were only present at the leading edge of the neo-epidermis. Similar to Ki67, only very few BrdU-positive cells were present in Mucomaix-based models. The length of re-epithelization at T + 1 week of the DED- and Matriderm-based models was larger than in the ex vivo human skin model (Figure 5C). The length of the re-epithelized area of the wound in DED- and MatriDerm-based models at T + 2 weeks, however, was comparable to that of the ex vivo human skin model. Mucomaix-based models, on the other hand, lacked the capacity to regenerate the burned epidermis. Thus, the DED- and MatriDerm-based models showed regenerative capacity with a neo-epidermis that contained proliferating cells, while regeneration in Mucomaix-based FSEs was not observed.

### 3.4. Cytokine Response of Burn-Injured Full Skin Equivalent Models

Cytokine response in the FSEs was explored by determining the cytokine levels in the culture medium at T0, T + 1–4 days, T + 5–7 days, and T + 8–11 days (Figure 6 and Appendix A). Of the 13 cytokines that were analyzed, high levels of IL-6, IL-8, and MCP-1 and moderate levels of IL-4 and IP-10 were found in both burn-injured and uninjured models. Only low levels of IL-12p70 and IFN-γ were detected in both burn-injured and uninjured models, while the levels of IL-2, IL-17A, and TNF-α were undetectable. The expression of IL-1β was the highest in ex vivo human skin models, and IL-10 expression appeared to be higher in both ex vivo human skin models and Mucomaix-based FSEs. TGF-β1, on the other hand, was more abundantly expressed in the FSEs than in the ex vivo human skin model.

In this explorative analysis, it seemed that the expression of IL-4, IL-6, IL-8, MCP-1, IFN-γ, and IL-12p70 was more or less consistent over time, as opposed to IL-1β, IL-10, IP-10, and TGF-β1. The levels IL-1β, IL-10, and IP-10 gradually decreased over time in the ex vivo human skin models, but this was not significant. In response to burn injury, the level of IL-4, IL-6, IL-10, and TGF-β1 showed a modest increase only in MatriDerm- and Mucomaix-based models at T + 1–4 days, although it did not reach significance. In the Mucomaix-based FSE, burn injury also increased the expression of IL-8, IL-12p70, and IFN-γ. Surprisingly, no differences were found for the ex vivo human skin model or DED-based FSE in reaction to the burn injury. Thus, high levels of pro-inflammatory cytokines were present in the medium of FSE models, similar to ex vivo human skin. The effect of burn injury on pro-inflammatory cytokines was limited and was only evident in the MatriDerm- and Mucomaix-based FSEs.

## 4. Discussion

Due to issues in the translation of animal data to the human situation, as well as ethical concerns, there is a growing demand for more appropriate, animal-free approaches in preclinical research [11]. Organotypic skin models, such as FSEs, are promising alternatives to animal models because they are more standardized, controllable, and easy to customize with relevant components such as specific cell types [20,21,33]. FSE models are more realistic than models that only use an epidermal layer, as the interplay between keratinocytes and fibroblasts affects skin development and healing [34,35,36]. Here, we validated FSEs generated from commercially available dermal substitutes MatriDerm and Mucomaix for the use as in vitro skin models to study skin development and burn wound healing. Additionally, we investigated the effect of longer culture times (up to a total of 5 weeks). Studies usually culture organotypic skin models up to 3 weeks [37,38,39,40], but in light of preclinical studies, culture times longer than 3 weeks could be required. In the development of 3D models, predominantly the histology, composition of the extracellular matrix, or cell survival is studied [41,42]. Our study not only showed that the FSEs were capable of forming a functional epidermis, but also showed that these models were able to regenerate after thermal injury.

Epidermal morphology of the FSE models after 3 weeks of culture was similar to that of ex vivo human skin. Normal epidermal differentiation was present during this period of culture, as shown by consistent expression of early and later differentiation markers cytokeratin 10 and involucrin. Extending the culture time by 1 or 2 weeks improved the organization of the epidermal structure and led to flattening of the stratum spinosum, as displayed by cytokeratin 10 expression, while the stratum corneum thickened. In FSE models, the expression of several markers of epidermal development was similar to the expression in ex vivo human skin and skin equivalent models from other researchers [20,21,43]. Involucrin was also present in the suprabasal layers of the FSEs, an observation similar to that of Coolen et al. and Thakoersing et al. [20,43]. Premature expression of involucrin is indicative of overactivated cell differentiation and is likely caused by an excess of growth factors in the culture medium or by an imbalance in the ratio of fibroblasts to keratinocytes [35,43,44]. With the extension of culture time, epidermal organization gradually improved, coinciding with the expression of collagen IV and laminin α 5. Despite a well-formed basement membrane in the MatriDerm- and Mucomaix-based FSEs, the basal layer was not entirely organized and did not improve over time. The disruption of the basal layer could be related to the porosity of the collagen matrices causing keratinocytes to partially descend into the matrix. In all models, a basement membrane was present, which was more mature at T + 2 weeks. All models contained a fibroblast-populated dermis, as was shown by vimentin expression [45].

Both MatriDerm- and Mucomaix-based models appeared suitable for in vitro study of skin development. However, due to the rapid degradation of Mucomaix, this matrix turned out less suitable as a model for extensive culture times. The degradation speed of Mucomaix was also shown in vivo by Udeabor et al. [28]. The MatriDerm matrix also degraded over time but clearly at a slower rate, which might be due to the presence of elastin-hydrolysate, making it less susceptible to enzymatic degradation [46]. Because of its faster degradation rate, Mucomaix could be more suitable for the study of degradation and cell matrix interactions. Clinically, Mucomaix is useful for the repair of intra- and extra-oral defects [28].

Burn injury and regeneration could successfully be studied in MatriDerm- and DED-based FSEs, as they displayed a regenerative and proliferative capacity similar to ex vivo human skin. The faster re-epithelization rate in FSEs at T + 1 week could be related to an increased proliferation in the FSEs due to the culture of cells. In contrast, cells in intact skin, especially keratinocytes, might be programmed more for differentiation rather than proliferation. Several other studies have used in vitro skin models to study the effects of burn injury, but they did not study the rate of re-epithelization [39,47,48].

Cytokines IL-6, IL-8, and MCP-1 were expressed in the FSE models at levels similar to those in ex vivo human skin. Apparently, there is already some degree of stress response in these models that is presumably triggered by the culturing of cells and in vitro skin development. This is supported by the abundant expression of stress marker cytokeratin 17 (Appendix A). Cytokeratin 17 was also expressed in ex vivo human skin models, but only after culture. Reports on the cytokine expression of cells in response to culture or skin development are very limited. This cytokine response is, however, important to take into account, because inflammation and skin regeneration can affect each other, thereby potentially delaying wound healing processes. Despite the potential presence of stress and subsequent cytokine responses during in vitro culture, an epidermis and dermis were successfully established in the presented models.

As seen only in ex vivo human skin cultures, the level of IL-1β, IL-10, and IP-10 gradually decreased over time. These cytokines might have been produced by immune cells, such as lymphocytes, that were residing in the ex vivo skin. With increasing culture time, cytokine production would then be reduced due to migration or depletion of these cells. Burn injury had only a limited effect on the level of cytokines and seemed to moderately increase the levels of IL-4, IL-6, IL-8, IL-10, and TGF-β1 in MatriDerm- and Mucomaix-based models early after injury. Possibly, the effect of burn injury was minimal because of the initially high levels in uninjured models. An increase in IL-8 in medium of burn-injured in vitro skin models was shown by Breetveld et al. and was only present early after injury (up to 4 days) [39]. A study from Schneider et al. showed an increase in the levels of IL-6 and IL-8 in similar models, also during the first week after injury [48]. The limited effect of burn injury on these models is likely caused by the absence of blood circulation and immune cells, which are well-known inducers of immune reactions. Because the thermal injury damaged a large portion of cells in these relatively small models, the potential response could only originate from the remaining viable cells. When the population of remaining cells is too small, the response will also be rather limited.

The current FSEs are useful for the study of tissue development and repair and for translational research without the use of animal models [33,49]. When fibroblasts and keratinocytes are kept in frozen stock, these models can be produced on demand, unlike ex vivo skin models, which depend on the availability of donor skin. FSEs are also advantageous because they are more standardized, can minimize donor variation, and are easily adjustable in terms of matrix, cell types, and cell numbers. 

The next step in the development of in vitro skin models will be the integration of immune cells, blood vessels, or other relevant skin appendages [50,51,52,53] and developing models suitable for drug discovery and testing [54]. Cells from different (disease-related) origins, such as skin cells derived from fetal, burn, or scar tissue, could be used to study their effect on skin regeneration. For example, van den Broek et al. developed a hypertrophic scar model using adipose-derived mesenchymal stem cells [55]. In these scar models, differences in contraction, epidermal thickness, and cytokine response were shown compared to models produced from dermal mesenchymal cells. To study inflammatory responses in a more relevant environment, immune cells can be integrated into FSEs [56]. Finally, such models could be supplemented with skin appendages such as hair follicles, making these models a useful platform to test interventions in the preclinical stage.

## 5. Conclusions

Clinically applied matrices MatriDerm and Mucomaix are suitable materials for in vitro skin model development. MatriDerm-based FSEs could be used for extensive culture periods and demonstrated regeneration after thermal wounding. The cytokine response of FSEs was comparable to that of ex vivo human skin. These models are therefore useful for the study of skin development and wound healing using a uniform dermal component without the need for animal models. Further development of the FSEs could include the addition of various immune cells, which would allow further study of inflammatory processes and testing of novel therapeutics.

## Figures and Tables

**Figure 1 jfb-14-00029-f001:**
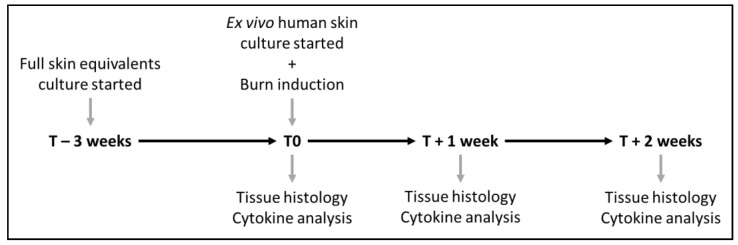
Experiment scheme showing the timing of the performed steps.

**Figure 2 jfb-14-00029-f002:**
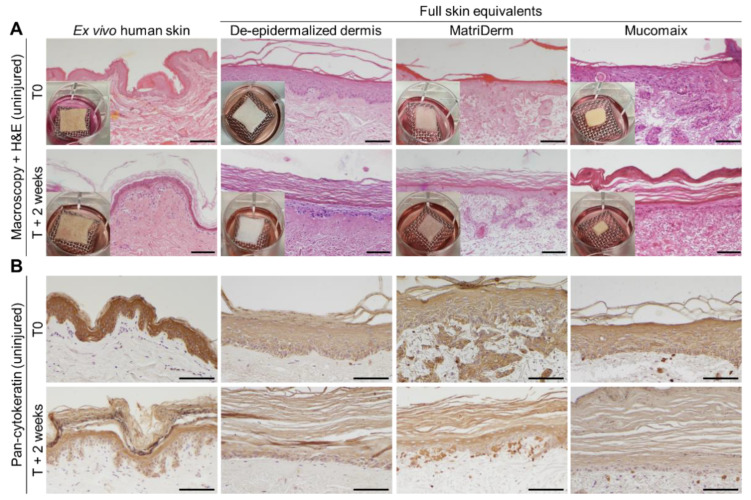
Evaluation of skin development of cultured skin models. Images of ex vivo human skin (**left**) and full skin equivalents generated from de-epidermalized dermis, MatriDerm, and Mucomaix (**right**) at T0 or T + 2 weeks. (**A**) Macroscopy and H&E staining; (**B**) Immunohistochemical pan-cytokeratin staining. Models were produced from 3 different skin donors in duplicate. For the full skin equivalent models, T0 was after the initial 3 weeks of culture. Black scale bar = 100 µm.

**Figure 3 jfb-14-00029-f003:**
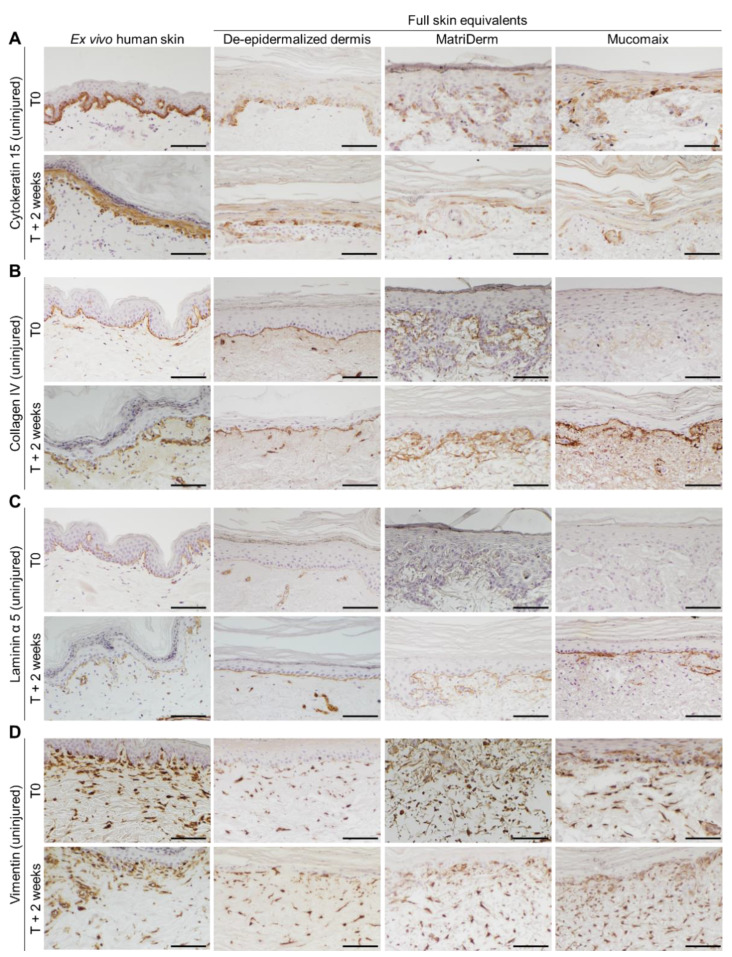
Evaluation of cytokeratin 15, collagen IV, laminin α 5, and vimentin expression in cultured skin models. Images of ex vivo human skin (**left**) and full skin equivalents generated from de-epidermalized dermis, MatriDerm, and Mucomaix (**right**). Immunohistochemical (**A**) cytokeratin 15, (**B**) collagen IV, (**C**) laminin α 5, and (**D**) vimentin DAB staining. Models were produced from 3 different skin donors in duplicate. For the full skin equivalent models, T0 was after the initial 3 weeks of culture. Black scale bar = 100 µm.

**Figure 4 jfb-14-00029-f004:**
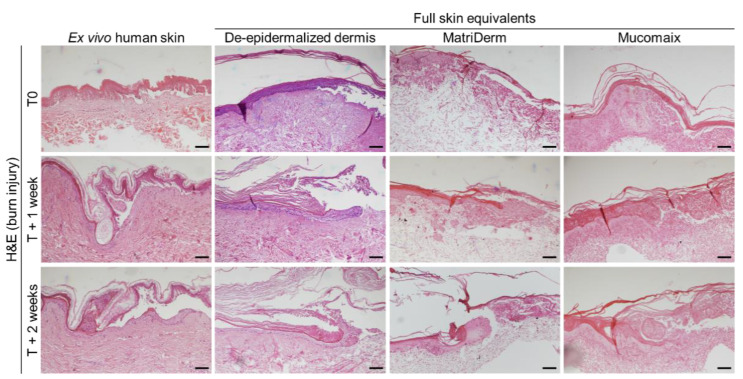
Effect of burn injury on cultured skin models. Images of ex vivo human skin (**left**) and full skin equivalents generated from de-epidermalized dermis, MatriDerm, and Mucomaix (**right**). Models were produced from 3 different skin donors in duplicate. For the full skin equivalent models, T0 was after the initial 3 weeks of culture. Black scale bar = 100 µm.

**Figure 5 jfb-14-00029-f005:**
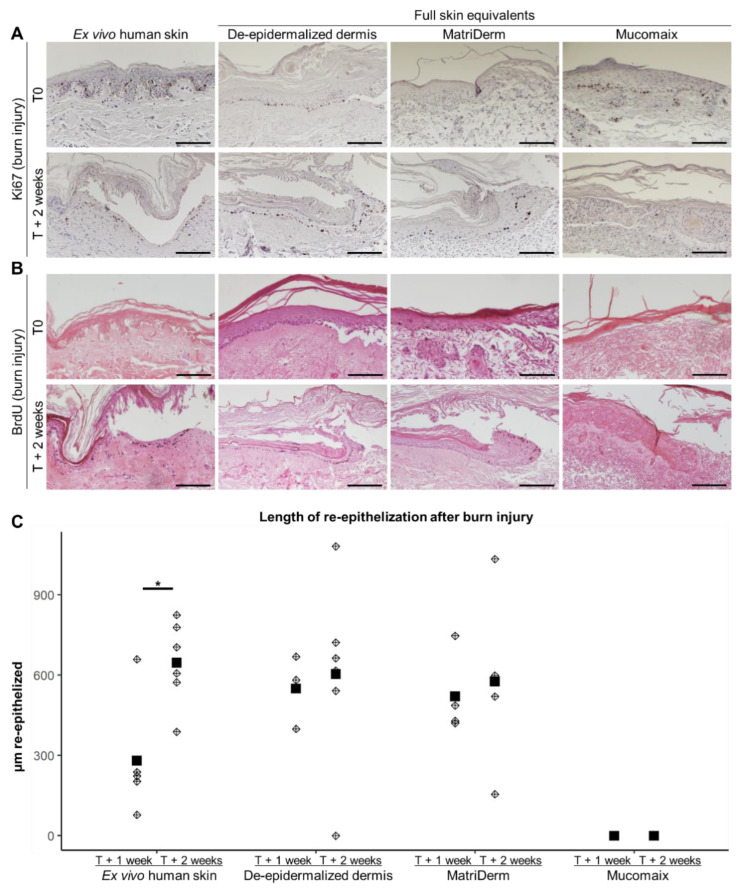
Evaluation of proliferation and re-epithelization in burn-injured skin models. Images of ex vivo human skin (**left**) and full skin equivalents generated from de-epidermalized dermis, MatriDerm, and Mucomaix (**right**) at T0 and T + 2 weeks after burn. Immunohistochemical (**A**) Ki67; (**B**) BrdU DAB staining. Because the culture of ex vivo human skin models started at T0 and BrdU needed to be added 24 h prior to termination of the models, no BrdU was present in these models at T0. (**C**) Length of re-epithelization after burn injury at T + 1 week and T + 2 weeks after burn (diamonds represent the mean per model, and squares represent the mean of all models). Models were produced from 3 different skin donors in duplicate. For the full skin equivalent models, T0 was after the initial 3 weeks of culture. Black scale bar = 100 µm. Statistically significant differences are indicated by asterisks: *: *p* < 0.05.

**Figure 6 jfb-14-00029-f006:**
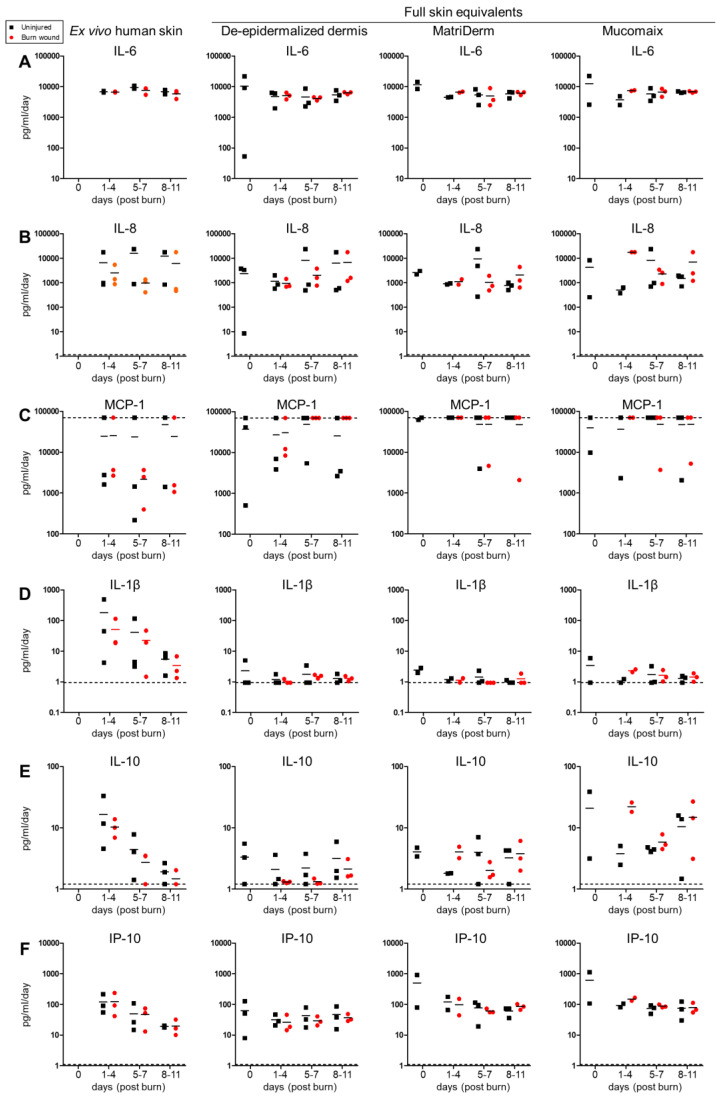
Cytokines detected in medium of burn-injured and uninjured skin models. Ex vivo human skin (**left**) and full skin equivalents generated from de-epidermalized dermis, MatriDerm, and Mucomaix (**right**). Level of (**A**) IL-6; (**B**) IL-8; (**C**) MCP-1; (**D**) IL-1β; (**E**) IL-10; (**F**) IP-10 in the culture medium at T0, T + 1–4 days, T + 5–7 days, and T + 8–11 days (after burn injury). Samples from biological duplicates were pooled per donor (n = 3 donors) and re-calculated into pg/mL per day of culture to compensate for differences in intervals of medium changes. Striped line indicates the highest or lowest level of quantification. Because ex vivo human skin models were started on the first day, no levels are shown for day 0.

## Data Availability

The raw data supporting the conclusions of this article will be made available by the authors, without undue reservation.

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
