# Peer review of "Full Skin Equivalent Models for Simulation of Burn Wound Healing, Exploring Skin Regeneration and Cytokine Response"

_jfb, 2023, doi:10.3390/jfb14010029_

Round 1

Reviewer 1 Report

1. What are the advantages of the proposed method for burn wound assessment in this paper compared to common (clinical) treatments? The comparison and the advantages can be added to the manuscript.

2. Figure 6, the manuscript lacks information on experimental replication. This is particularly worrisome. Please revise the manuscript detailing your experimental and technical replications.

3. Compared to commonly used wound dressings (such as hydrogel, 10.1021/acsmacrolett.2c00290), what is the most important feature of the current simulation system?

4. There are some formatting errors in the article. For example, spelling of references must be checked to meet the journal style (Reference 32). Please check carefully and use it properly.

Author Response

We would like to thank the reviewer for her/his comments and respond to these comments point by point:

  1. The proposed in vitro models are not meant to replace (clinical) treatments. We have developed these models to study wound healing at a more basic/fundamental level without the need of animals. Moreover, the models can be used to study the effect of pharmaceuticals on wound healing prior to clinical testing. We have added some sentences to the Introduction and Discussion section about the advantages of the proposed burn wound model for these purposes.
  2. The reviewer has a fair point and we have made this more clear in the legend of Figure 6 and in the Materials and Methods section.
  3. Our full skin equivalents have several advantages over wound dressings such as hydrogels:
  • Skin cells can be easily cultured in and on our 3D scaffolds, whereas this is not the case in wound dressing materials.
  • Skin cells will experience a ‘natural’ environment, similar to normal cells, as the scaffolds we used contain collagen as a predominant component. Thus, skin fibroblasts will produce a.o. collagen, as indicated in our manuscript.
  1. We have completely checked and revised our manuscript to correct errors and increase readability. We have adjusted the reference list to fully adhere to the citation style of the journal.

Reviewer 2 Report

Dear authors

Thank you so much for your submission. However, I would like to suggest you to improve introduction more on current research and clinical trials.

in figure 2, 3 and 4, pointed the changes on tissues

Author Response

We would like to thank the reviewer for reviewing our manuscript. Our manuscript was fully checked for errors and the English language was improved to increase readability. We have revised the Introduction section and have included references to several papers describing similar models. Moreover, we added studies that describe the use of MatriDerm and Mucomaix in in vivo and clinical settings to motivate our choice of dermal matrices. We anticipate that the proposed models are useful for the in vitro study of skin regeneration. The use of these models or similar models in clinical trials is however not within the scope of this paper.

Since in Figures 2-4 many differences can be noted between the different tissues, it is not feasible to indicate differences in the figures, as this would hamper the visibility. Therefore, we described the differences in the text of the Results section.

Reviewer 3 Report

The work is interesting and develops new in vitro FSEs models. However, I have a few critical remarks and comments. The title of the paper suggests that the model of inflammation will also be studied. I suggest removing this part of the title because the paper does not analyze typical inflammation in the skin. Apoptosis or necrosis was not assessed in the study, which may be an important element of the interpretation of the results. Minor comments: figure 3, incorrect order of legends in figures c) and d). I have not found information about the total number of patients from whom the material was obtained and their clinical characteristics. Figure 5: it would be advisable to show the staining also at time 0 (control). In my opinion, the work is worth publishing after taking into account the above comments.

Author Response

We would like to thank the reviewer for her/his comments and agree with the suggestion to change the last part of the title. We have changed the title to “Full Skin Equivalent Models for Simulation of Burn Wound Healing, Exploring Skin Regeneration and Cytokine Response” to be more specific about the aspects of inflammation that were studied in this paper.

We appreciate the comment from the Reviewer about apoptosis and necrosis. Indeed, the burn injury applied in some of these models does induce a certain level of necrosis (as shown in our earlier work of Coolen et al. from 2008). However, in this study, we focused on regeneration of epidermis and dermis and effects on inflammatory mediators. Viability of these models was shown by Ki67, BrdU and several cytokines, even after a total of 5 weeks of culture. Nevertheless, it would be interesting to include further analysis of apoptosis and necrosis in our future work.

The reviewer is right about the minor error in the legend of Figure 3 and we have changed this accordingly. We have added the following information to the Materials and Methods section: “Eleven skin tissue samples were used, originating from abdominal, leg or arm reconstructions wherein excess skin was removed (donor age: 43.8±11.7 years old; donor sex: 72.7% female)”. Furthermore, we have added time point 0 (T0) to Figure 5.

Reviewer 4 Report

In this work, the authors studied the Full Skin Equivalent Models for the Simulation of Burn Wound Healing and Inflammation. They generated FSEs from clinically approved dermal substitutes MatriDerm® and Mucomaix®. They validated these models by studying the skin development, cell differentiation, cytokine production, and viability, and compared their performance to cultured ex vivo human skin and de-epidermalized-dermis (DED) based FSEs. Also, they achieved good results and their findings are noticeable. But there are some points that the authors should consider to improve the quality of their work.

1- The introduction didn't provide sufficient background.

2- The introduction didn't include all relevant references.

3- The conclusion should be extended.

4- in lines 394 and 395 "This has already been demonstrated by van den Broek et al. for modeling of hypertrophic skin cells [41]". need more description.

5- The aim of this study is not mentioned in the abstract section.

6- the manuscript needs grammatical and spelling corrections.

7- why the authors didn't mention any things about ethical concerns in this study?

Author Response

We would like to thank the reviewer for her/his comments and respond to these comments point by point:

  1. We have extended our Introduction section to provide more background about the study.
  2. In the new and extended version of our Introduction additional references (such as Hosseini et al. 2022; Ozdogan et al. 2021; Liu et al. 2022 and Urciuolo et al. 2022) were included and discussed concerning similar models as well as studies that describe the use of MatriDerm and Mucomaix in in vivo and clinical settings to motivate our choice of dermal matrices.
  3. The Conclusion section was extended and now includes the following sentences: “Clinically applied matrices MatriDerm and Mucomaix are suitable materials for in vitro skin model development. MatriDerm-based FSEs could be used for extensive culture periods and demonstrated regeneration after thermal wounding. The cytokine response of FSEs was comparable to that of ex vivo human skin. These models are therefore useful for the study of skin development and wound healing using a uniform dermal component without the need for animal models. Further development of the FSEs could include addition of various immune cells, which would allow further study of inflammatory processes and testing of novel therapeutics.”
  4. More description was added to this part of the Discussion: “For example, van den Broek et al. developed a hypertrophic scar model using adipose-derived mesenchymal stem cells [55]. In these scar models, differences in contraction, epidermal thickness and cytokine response were shown compared to models produced from dermal mesenchymal cells.”
  5. We added the aim of the study to the Abstract: “Our aim was to validate the MatriDerm- and Mucomaix-based FSEs for the use as in vitro models of wound healing. Therefore, we first characterized the FSEs in terms of skin development and cell proliferation.”
  6. The complete manuscript was revised to increase readability and grammar and spelling corrections were made.
  7. The institutional informed consent procedure was included in our Materials and Methods section. In the revised version of our manuscript, we have elaborated on this procedure and we have sent the patient information form that was used for this study to the editor.